# Design and Analysis of Solid Rocket Composite Motor Case Connector Using Finite Element Method

**DOI:** 10.3390/polym14132596

**Published:** 2022-06-27

**Authors:** Lvtao Zhu, Jiayi Wang, Wei Shen, Lifeng Chen, Chengyan Zhu

**Affiliations:** 1College of Textile Science and Engineering (International Institute of Silk), Zhejiang Sci-Tech University, Hangzhou 310018, China; wjoy18867578703@163.com (J.W.); cyzhu@zstu.edu.cn (C.Z.); 2Shaoxing-Keqiao Institute, Zhejiang Sci-Tech University, Shaoxing 312000, China; 3Shaoxing Baojing Composite Materials Co., Ltd., Shaoxing 312000, China; shenw@jinggonggroup.com (W.S.); vincentchenli@sina.com (L.C.)

**Keywords:** solid rocket motor case (SRMC) connector, carbon fiber, lay-up, mechanical properties, finite element

## Abstract

The connector is an essential component in the solid rocket motor case (SRMC), and its weight and performance can directly affect the blasting performance of SRMC. Considering the lightweight design of these structures, fiber-reinforced composite materials are used for the major components. In this study, the finite element analysis of the SRMC connector was performed. The lay-up design and structure optimum design of the connector were studied. Furthermore, the strain distribution on the composite body was compared with experimental measurements. The results demonstrate that the calculated value of the final preferred solution was within the allowable range, and at least 31% weight loss was achieved, suggesting that the performance of the optimum design was optimized. The comparison between the finite element calculation and the test results suggests that the design was within the allowable range and reasonable.

## 1. Introduction

Composite materials can be classified according to the type of strengthening material into particle-reinforced and fiber-reinforced composite materials. Furthermore, fiber-reinforced composite materials can be divided into short fiber-reinforced and long fiber-reinforced composite materials [1]. A solid rocket composite shell is a kind of long fiber-reinforced composite material, which is pre-impregnated with carbon fiber (or glass fiber) and wound around the core mold layer by layer, before solidification at a certain temperature. The solid rocket motor case (SRMC) is mainly composed of a tube structure, head, insulation layer, and skirt, and it has been widely applied in space vehicles, missile weapons, and other fields as a crucial part of the rocket motor. An SRMC connector is employed to connect the engine nozzle and ignition device, as shown in Figure 1. The design of the case connector structure significantly contributes to the development of the composite case. Moreover, its performance directly affects the blasting performance of the composite case. Due to their light weight, high strength, and high stiffness, fiber-reinforced composites can efficiently decrease the structure mass of the SRMC and increase the range of the rocket, providing great military and economic benefits [2]. For example, the range of a strategic missile can be increased by 16 km if the mass of the third structure of the solid rocket engine is reduced by 1 kg [3].

At present, carbon fiber-reinforced composites are widely used for strategic missiles and delivery systems, with several related papers published in this field. Ramanjaneyulu et al. [4] explored the SRMC using the finite element method, revealing that the hoop stress was gradually increased from the outer layer to the inner layer in all parts of the SRMC. Özaslan et al. [5] designed and analyzed a filament wound composite SRMC with finite element analysis and compared burst tests regarding the fiber direction strain distribution through the outer surface of the motor case to verify the analysis. Niharika et al. [6] used the simulation software ANSYS (R 18.0, ANSYS Inc., Canonsburg, PA, USA) to design a composite rocket motor casing. Hossam et al. [7] proposed that filament winding was the best technique for the production of composite pressure vessels (CPVs) in a short time, and different materials (including conventional and composite materials) were suitable for the design of SRMC structures. They also studied and summarized the optimum design of SRMC structures. Shaheen et al. [8] developed a 3D model of SRMC using CATIA V5R16 software (V5R16, Dassault Systems, Waltham, MA, USA) and conducted static structural analysis and linear buckling analysis for different stack-ups of a unidirectional carbon–epoxy composite and D6AC steel material rocket motor casing to specify the more efficient material. Prakash et al. [9] successfully designed and developed VEGA SRMC and discussed the effect of material mismatch on the static behavior of the flex seal, which contributed imperatively to the development of composite rocket motor casings.

A large number of studies related to CERP have been carried out in other sectors. Juan [10] accurately modeled the winding layer of composite pressure vessels using the fiber winding pressure vessel plug-in WCM, as well as carried out a finite element simulation and blasting test verification on a type IV high-pressure hydrogen storage cylinder with the designed pressure of 70 MPa. Johansen et al. [11] designed a fiber winding analysis program and realized the winding analysis of any axisymmetric rotating body and its combination through an integrated CAD/CAE/CAM design method. Ambach [12] combined CFRP with steel and applied it to the manufacturing of an automobile roof, revealing that the mechanical properties of the material achieved good performance in terms of the crushing resistance of the automobile roof. Wang [13] explored the use of carbon fiber composite materials in biomedical science. Using barium titanate–hydroxyapatite (BT–HA) composite material as the matrix, Cf/BT–HA composite material was prepared to improve the artificial bone due to poor mechanical properties. Liang [14] studied the application of carbon fiber composite materials in bogies of rail transit vehicles, considering the properties of carbon fiber composite materials, such as high strength, high toughness, fatigue resistance, high temperature resistance, corrosion resistance, and light weight; he proposed a rectification plan for the use of carbon fiber composite material as a safety support in current vehicles. In terms of the spinning process, Kovarskii et al. [15] analyzed the structure of carbon fibers such as T800HB using EPR spectroscopy and X-ray diffraction, and they found that the microstructure of carbon fibers is directly related to their mechanical properties.

To date, many theoretical models related to SRMCs have been reported [16,17,18,19]. However, research on SRMC connectors is still insufficient. Generally, the case connector is the main force component, and the loading condition is complex. Meanwhile, the case connector is extremely sensitive to internal imperfections, necessitating methods to effectively improve its mechanical properties and dimensional accuracy. As is known, the SRMC connector operates in a high-temperature environment. Although the connector’s external layer is protected by an insulating layer, the surface temperature can still reach up to a maximum of 250 °C. Furthermore, the dimensional stability of the connector is another basic item, and titanium alloy with excellent comprehensive properties has been broadly used in the connector. However, titanium alloy is expensive with high density, making it a single component with a large mass. The metal connector accounts for more than 15% of the total mass of the case. Furthermore, the process of manufacturing the metal connector is rather complex with a long cycle and high cost. Thus, it is urgent to develop a new material that can replace the metal material in the SRMC connector.

In this study, finite element analysis of the SRMC connector was performed. The lay-up and optimum structure designs of the connector were exported. The FEM simulation results were shown to be similar to experimental results. Thus, the performance of the optimum design was successfully improved.

## 2. Experimental Analysis

The SRMC connector RS05A was made by using the mold pressing process with carbon fiber T300 fabric prepreg (Toray Inc., Lacq, France). The thickness and density of the single layer were 0.235 mm and 1.55 g/cm^3^, respectively. Toray T700SC (12K, Toray Inc., Lacq, France)) carbon fiber was employed to produce a motor case with fiber winding technology. The fiber winding shell of the solid rocket motor was made of T700 fiber/epoxy resin composite material (Kosan Inc., Tokyo, Japan),, with a resin content of 32% and fiber content of 150 g, a viscosity of 300 mPa·s at room temperature, an opening period of 8–12 h, a curing temperature of 150 °C for 4 h, and a glass transition temperature of 170 °C. The bushing was embedded in the composite made of aluminum (AL7075-T6, Moju Inc., Shanghai, China) material, as shown in Figure 2b. The mechanical properties of carbon fiber, P700-1M resin, and AL are presented in Table 1.

## 3. Structural Design

The primary connector structure of the RS05A front connector is illustrated in Figure 1a. After the first round of optimization using the finite element method, the length of the connector was adjusted to make the street longer, so as to fit more closely with the shell wall, as exhibited in Figure 1b. The primary front connector was 10 mm from the opening cut, while the optimized one was 20 mm from the opening cut.

Since the connector was fixed on the case using bolts, the primary RS05A metal connector was designed with M28 × 1.5 threaded hole in the middle. The bushing made of AL-7075-T6 was embedded in the composite connector to solve the problem of the composite being difficult to use as the metal connector.

According to the actual prepreg lay-up effect, the RS05A composite connector was designed to reduce the number of multilayer step structures during the production process, facilitating the insertion of the prepreg in the step structures. Meanwhile, the AL bushing embedded in the composite structure was adjusted into trapezoidal modes in order to ensure the flatness of the inner surface of the case and the thickness of the bushing root. In this way, an optimal design could be achieved.

## 4. RS05A Lay-Up Mode

Considering the operability of the actual production process, the final optimized lay-up is illustrated in Figure 3a. The optimized lamination of the composite was cured by a secondary co-curing process. The lamination was laid on the core die, and the lower half of the connector was cured by a molding process after lamination was completed. The secondary co-curing treatment was conducted when the top end face of the connector was laid up again. After curing, the connector was machined to the theoretical shape. Finally, the intermediate insert was embedded into the composite connector body using an adhesive.

The material used for the composite connector was carbon fiber T300 biaxial fabric, which was spread as an isotropic material in the form of a patchwork butt. The prepreg layering table of the connector structure is shown in Table 2. Given the large radian shape of the front connector, it was necessary to shear the pavement. The cutting opening mode is presented in Figure 3b.

## 5. Finite Element Model

The shell layer was designed by grid theory [20], and the designed burst pressure was 15 MPa. The finite element computer software, ABAQUS (V6.13, Dassault Systems, Waltham, MA, USA), was employed for SRMC burst pressure simulation. The dimensions of the finite element model were the actual dimensions. The SRMC and connector were meshed with linear reduced integration solid elements (C3D8R), with a mesh size of approximately 10 mm. The model had a total of 23,355 cells and 29,020 nodes. Table 3 presents the front connector weight of different schemes. It can be seen that the front connector weight of the initial plan was 0.41 kg, while that of the optimized scheme was 0.408 kg; the corresponding weights of the AL insert were 0.056 kg and 0.0613 kg, respectively. The percentage weight loss of the optimized plan, final scheme, and experimental measurement was 31.4%, 31.0%, and 30.6%, respectively.

### 5.1. Lay-Up Information Table

The lay-up of the front connector of the RS05A composite was quasi-isotropic, the lay-up of the RS05A composite connector was divided into five directions (0°, 22.5°, 45°, 67.5°, and 90°), and the lay-up ratio was 1. In the actual lay-up, each layer was rotated by a certain angle to disperse the lap position and angle.

### 5.2. Loading and Constraints

Abaqus was used for linear loading calculation to achieve progressive failure analysis. In the finite element analysis, at each incremental step, the first equilibrium equation was solved, and the stress and strain of each layer of the element covered the stress and strain of the previous step. According to the damage mode, the stiffness could be reduced by changing the material parameters of the integral point. The equilibrium equation was reestablished, and the next load increment step was substituted. If the structure relative stiffness value of the current load step (the ratio of the current stiffness to the initial stiffness) tended to zero and began to soften and enter the unloading state, the structure was considered to have lost the bearing capacity, necessitating the progressive failure analysis of the wound shell [21].

The RS05A Motor Case mainly bears internal pressure. Axial displacement constraints were applied to the middle part of the shell to avoid rigid body displacement in the finite element calculation, and cyclic symmetry conditions were applied to the sides of the shell and joint model. Uniformly distributed pressure was applied on the inner surface of the shell, increasing from 0 to 15 MPa. The boundary conditions for the SRMC and connector in ABAQUS are defined below. The *X*-axis translation of the RS05A motor case was constrained, in addition to the Y-direction translation of the upper and lower surface elements and the Z-direction translation of the front and rear surface elements. In other words, symmetric constraints were imposed on the motor case. The constraints are illustrated in Figure 4.

The load condition was 15 MPa of internal blasting press.

Contact: Since the front connector and the insert are two parts, there may be relative friction between them. In this study, the contact constraint conditions were imposed on the bottom end face, the side of the front connector, and the bushing. The friction form was set using a friction coefficient of 1.5.

The SRMC failure criteria proposed by Hashin criteria [22] were applied to detect the failure modes in the fiber and matrix under both tension and compression failures, which involve four failure modes. The failure modes included in Hashin’s criteria are expressed below.

Tensile fiber failure for $\sigma_{11}$ ≥ 0:(1)$${\left(\frac{\sigma_{11}}{X_{T}}\right)}^{2}+\frac{\sigma_{12}^{2}+\sigma_{13}^{2}}{S_{12}^{2}}\;\geq \;1.$$

Compressive fiber failure for $\sigma_{11}$ < 0:(2)$${\left(\frac{\sigma_{11}}{X_{c}}\right)}^{2}\;\geq \;1.$$

Tensile matrix failure for $\sigma_{22}$ + $\sigma_{33}$ > 0:(3)$$\frac{{\left(\sigma_{22}+\sigma_{33}\right)}^{2}}{Y_{T}^{2}}+\frac{\sigma_{23}^{2}-\sigma_{22}\sigma_{33}}{S_{23}^{2}}+\frac{\sigma_{12}^{2}+\sigma_{13}^{2}}{S_{12}^{2}}\;\geq \;1.$$

Compressive matrix failure for $\sigma_{22}$ + $\sigma_{33}$ < 0:(4)$$\left[{\left(\frac{Y_{C}}{2S_{23}}\right)}^{2}-1\right]\left(\frac{\sigma_{22}+\sigma_{33}}{Y_{C}}\right)+\frac{{\left(\sigma_{22}+\sigma_{33}\right)}^{2}}{4S_{23}^{2}}+\frac{\sigma_{23}^{2}-\sigma_{22}\sigma_{23}}{S_{23}^{2}}+\frac{\sigma_{12}^{2}+\sigma_{13}^{2}}{S_{12}^{2}}\;\geq \;1.$$

Interlaminar tensile failure for $\sigma_{33}$ > 0:(5)$${\left(\frac{\sigma_{33}}{Z_{T}}\right)}^{2}\;\geq \;1.$$
(6)$${\left(\frac{\sigma_{33}}{Z_{C}}\right)}^{2}\;\geq \;1.$$

Here, the $\sigma_{ij}$ terms are components of the stress tensor, i and j are local coordinate axes parallel and transverse to the fibers in each ply, respectively, and the *z*-axis coincides with the through-thickness direction.

Statical analysis using FEM was performed for the RS05A Motor Case, where the connector received complicated stress under high internal pressure. The mechanical responses and damage morphology of the FE models were obtained.

## 6. Results and Discussion

### 6.1. Analysis Results of the RS05A Front Connector

Pressure was applied on the shell; then, the shell was enlarged and deformation occurred in the middle of the front connector. This phenomenon was due to the existence of the pressure exerted internally. The deformation and maximum shear stress diagrams of the front connector under 15 MPa of blasting pressure are exhibited in Figure 5. It can be seen that the magnitude deformation of the front connector reached 5.376 mm. The maximum shear stress in the *XY*-direction was 2.57 MPa. Table 4 presents the displacement and shear stress results of the design.

The stress distribution of the front connector is presented in Figure 6 and Figure 7. As can be seen, the maximal tensile and compressive stress calculated using FEM was 492 MPa and −537 MPa, respectively. As shown in Table 4, the FEM results and experimental measurements were in agreement with the practical values.

Figure 8 shows that the Von Mises stress of AL inserts was 332.4MPa. As shown in Table 4, the stress–strain values obtained from the simulations were all within the permissible limits obtained from the experiments.

### 6.2. Experimental Results

A water pressure blasting experiment is designed for the shell to monitor the strain displacement change of the shell during blasting. Strain monitoring points were uniformly set on the shell and front connector, as shown in Figure 9. Deformation was relatively larger in the process of the booster, with resin shell cracking. Due to some damage of the strain gauge, the strain value could not be displayed. The complete results of the test points were generated, with each strain measuring point monitoring strain changes in both directions. The strain of point 8 at the small polar hole generated a sudden change. Two points were set for displacement change monitoring, which coincided with strain monitoring points 2 and 5. The results of extracting the two point shifts are shown in Figure 10. When the pressure reached 33 MPa, displacement occurred at both points. Combined with the strain displacement test results in the experiment, it can be seen that the actual burst pressure was 33 MPa. The hydrostatic test showed that the cylinder could meet the internal pressure of 15 MPa in working conditions and 33 MPa in blasting conditions.

The calculation results of typical schemes are summarized in Table 4. As shown, the experimental measurements, such as the front connector deformation, tensile and compressive stress in the *XY*-direction, shear stress in *XY*-plane, and Von Mises stress of AL inserts, were preferred as the final solution.

The winding angle used in the calculation (18.5°) was the average winding angle, while the actual winding angle at the equator was about 26°, i.e., the winding angle from the middle part of the barrel to the equator of the back head changed from 18.5° to 26°, resulting in an increase in the actual torsional stiffness of the cylinder near the back head. Therefore, the measured circumferential strain value was small. In the actual working condition, the stress of the shell would be better than that of the proposed design, and no damage would occur with the calculated value. Both calculated stresses were within the range of allowable design values. The design scheme and calculation met the requirements, and the design proposal was reasonable.

## 7. Conclusions


In this study, finite element analysis of the SRMC connector was performed. The lay-up and optimum structure designs of the connector were investigated. An experimental design was established, and the FEM simulation value was calculated. Loading and constrains were implemented in the FEM model. The actual experimental measurements were studied for a comparison. A blasting experiment was conducted to verify the simulation results.The maximum shear stress in the *XY*-direction was 2.57 MPa, the maximal tensile and compressive stress calculated using FEM was 492 MPa and −537 MPa, respectively, and the Von Mises stress of the AL insert was 332.4 MPa. The stress–strain values obtained from the simulations were all within the permissible limits obtained from the experiments.The results revealed that the calculated value of the final preferred solution was within the allowable range (Table 4), and at least 31% weight loss could be achieved (Table 2). This confirms that the performance of the optimum design was successfully improved.The accuracy of the modeling method was verified by analyzing the displacement and blasting pressure of the finite element simulation results. The comparison results showed that the FME result of blasting was 15 MPa, while the actual blasting was 33 MPa, suggesting that the simulated shell could meet the internal pressure in working conditions.


## Figures and Tables

**Figure 1 polymers-14-02596-f001:**
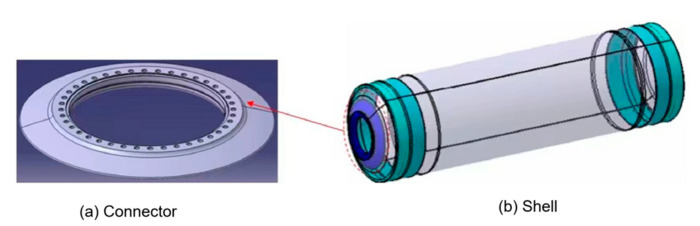
Schematic of SRMC connector and shell: (**a**) the connector to the shell body (**b**) the shell body of the rocket.

**Figure 2 polymers-14-02596-f002:**
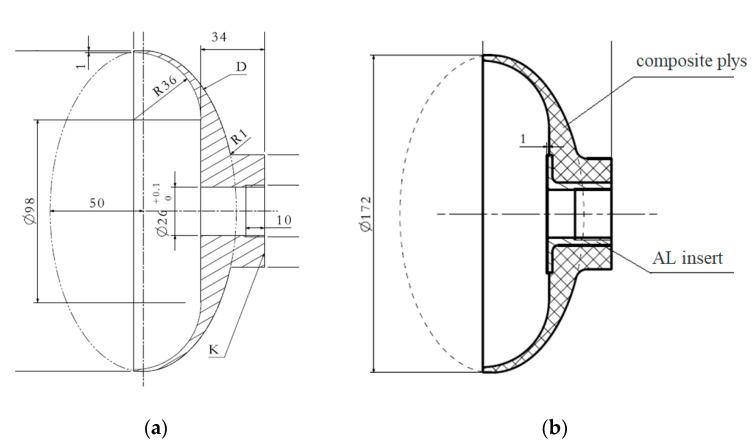
Profile of the RS05A front connector: (**a**) primary front connector (**b**) after first-round optimization.

**Figure 3 polymers-14-02596-f003:**
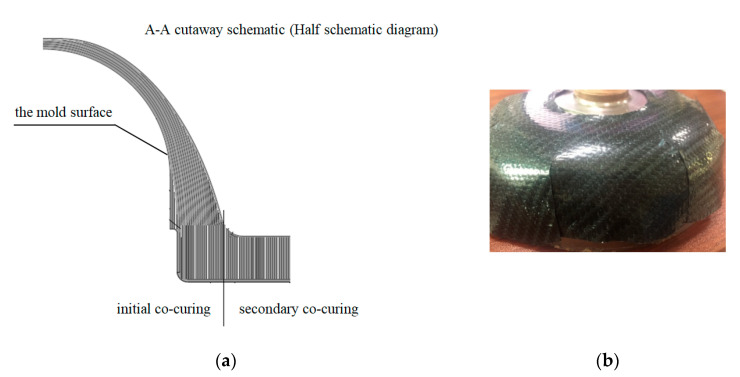
Lay-up diagram of the preferred scheme: (**a**) lay-up diagram (**b**) cutting opening mode.

**Figure 4 polymers-14-02596-f004:**
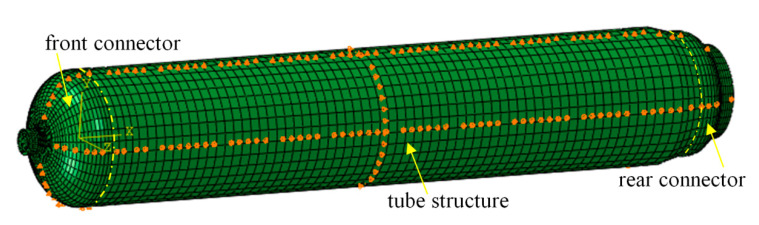
Schematic diagram of the RS05A motor case constraint.

**Figure 5 polymers-14-02596-f005:**
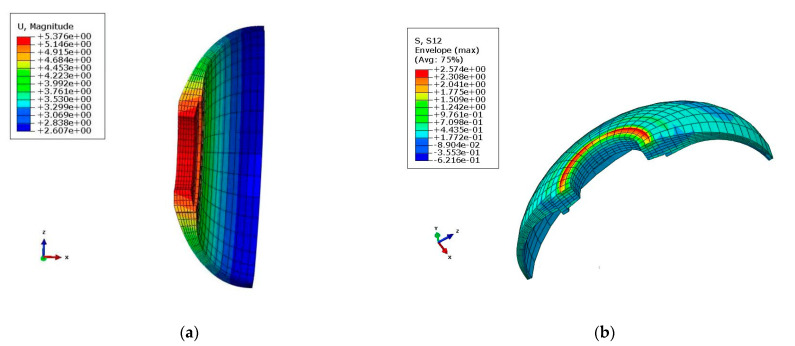
Deformation and maximum shear stress diagrams of the front connector: (**a**) deformation diagram; (**b**) maximum shear stress in *XY*-direction (τ_xy_ = 257 × 10^−^^2^ MPa).

**Figure 6 polymers-14-02596-f006:**
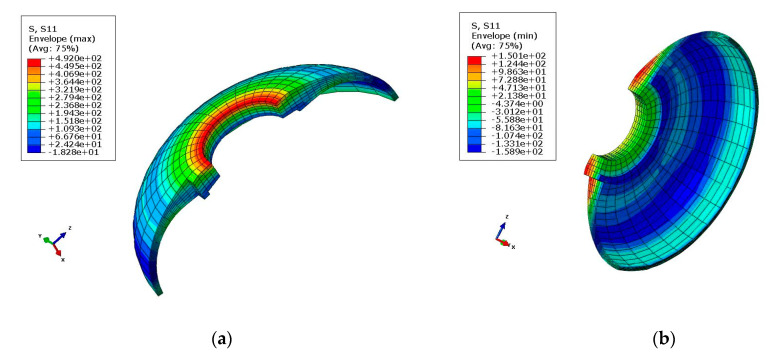
Maximal tensile and compressive stress diagrams in the *X*-direction: (**a**) Tensile stress (σ_x_ = 4.92 × 10^2^ MPa); (**b**) Compressive stress (σ_x_ = −1.589 × 10^2^ MPa).

**Figure 7 polymers-14-02596-f007:**
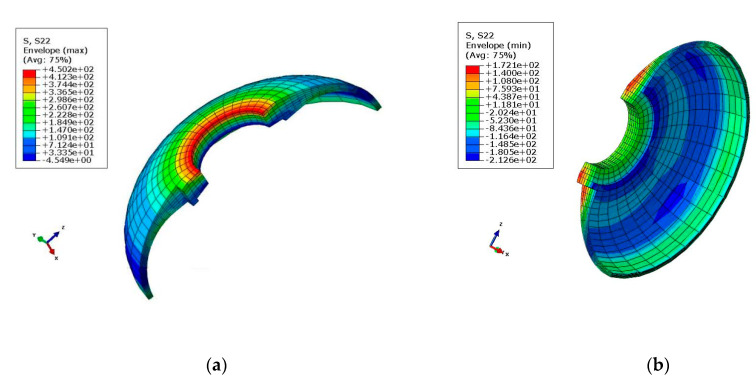
Maximal tensile and compression stress diagram in the *Y*-direction: (**a**) Tensile stress (σ_y_ = 4.502 × 10^2^ MPa); (**b**) Compression stress (σ_y_ = −2.126 × 10^2^ MPa).

**Figure 8 polymers-14-02596-f008:**
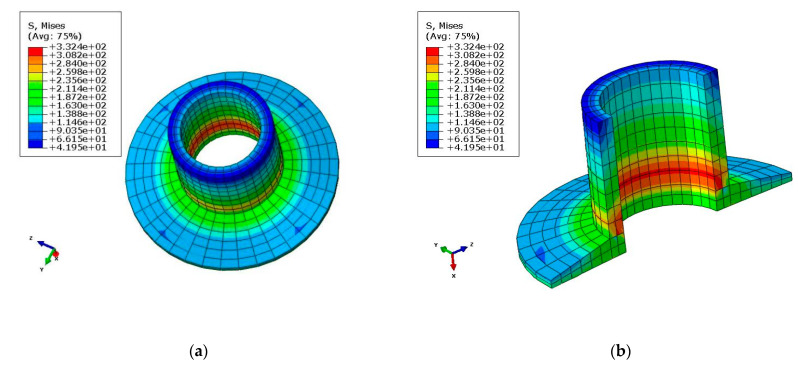
Von Mises stress of AL inserts (σ_y_ = 332.4 MPa): (**a**) Von Mises stress (overall view); (**b**) Von Mises stress (partial view).

**Figure 9 polymers-14-02596-f009:**
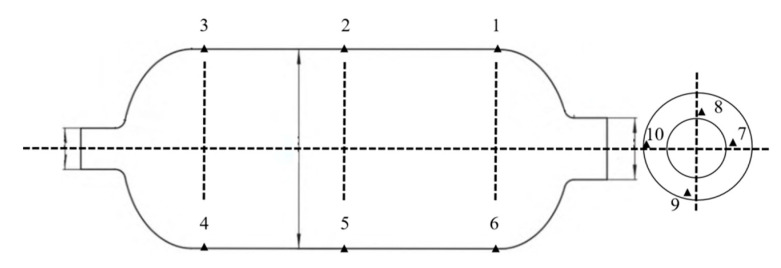
Stress measurement point.

**Figure 10 polymers-14-02596-f010:**
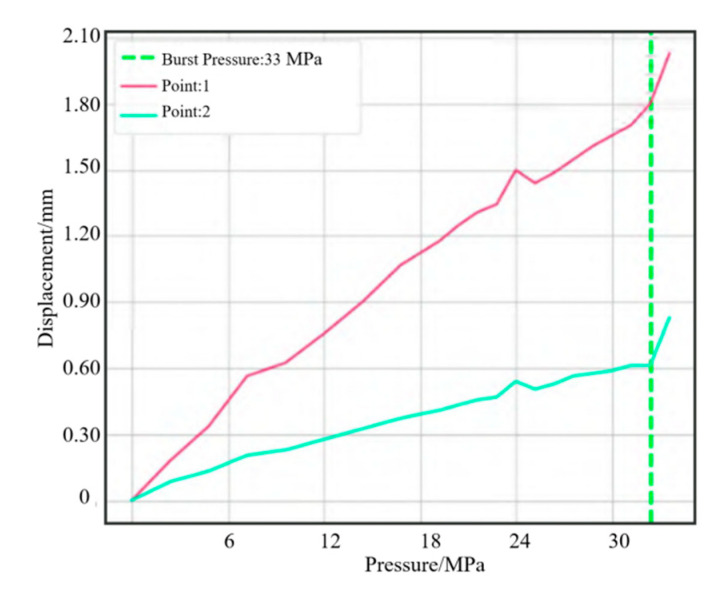
Tendency of displacement.

**Table 1 polymers-14-02596-t001:** Mechanical properties of the materials.

Property Items		Lamina Property	Resin	AL7075-T6
**Tensile strength (MPa)**	0°	665	70	570
	90°	552		
**Tensile modulus (GPa)**	0°	55.7	3.6	72
	90°	56.3		
**Compression strength (MPa)**	0°	500		
	90°	500		
**Compression modulus (GPa)**	0°	55.7		
	90°	56.3		

**Table 2 polymers-14-02596-t002:** The layering table of the connector structure.

Serial Number	Lay Up	Thickness (mm)	Angle (°)	Layer Number
1	Twill weaves of carbon fiber T300	0.225	0/90	1001
2	Twill weaves of carbon fiber T300	0.225	45/−45	1002
3	Twill weaves of carbon fiber T300	0.225	0/90	1003
4	Twill weaves of carbon fiber T300	0.225	45/−45	1004
5	Twill weaves of carbon fiber T300	0.225	0/90	1005
6	Twill weaves of carbon fiber T300	0.225	45/−45	1006
…	…	…	…	…
163	Twill weaves of carbon fiber T300	0.225	0/90	1163
164	Twill weaves of carbon fiber T300	0.225	45/−45	1164
165	Prepreg of carbon fiber T300	0.145	0	2001
166	Prepreg of carbon fiber T300	0.145	45	2002
167	Prepreg of carbon fiber T300	0.145	−45	2003
168	Prepreg of carbon fiber T300	0.145	90	2004
169	Prepreg of carbon fiber T300	0.145	0	2005
170	Prepreg of carbon fiber T300	0.145	45	2006
171	Prepreg of carbon fiber T300	0.145	−45	2007
172	Prepreg of carbon fiber T300	0.145	90	2008
…	…	…	…	…
285	Prepreg of carbon fiber T300	0.145	0	2021
286	Prepreg of carbon fiber T300	0.145	45	2122
287	Prepreg of carbon fiber T300	0.145	−45	2123
288	Prepreg of carbon fiber T300	0.145	90	2124
289	Prepreg of carbon fiber T300	0.145	0	2125
Total thickness	55.025 mm			

**Table 3 polymers-14-02596-t003:** Front connector weight of different schemes.

Location	AL Front Connector Scheme	The Initial Configuration	Optimized Scheme	Experimental Measurements
**Front connector weight (kg)**	0.68	0.41	0.408	0.411
**AL insert weight (kg)**	-	0.056	0.0613	0.0613
**Total weight (kg)**	0.68	0.466	0.469	0.472
**Percentage weight loss (%)**	-	31.4%	31.0%	30.6%

**Table 4 polymers-14-02596-t004:** Finite element calculation results.

		Final Preferred Solution		Experimental Measurements
	Allowable Value [23,24]	Calculated Value	Safety Factor	Experimental Value
**Front connector deformation (mm)**	-	1.92	-	1.98
**Tensile stress in *X*-direction (MPa)**	500	492.0	1.24	496.4
**Compressive stress in *X*-direction (MPa)**	−665	−158.9	1.02	−172.8
**Tensile stress in *Y*-direction (MPa)**	552	450.2	1.23	463.6
**Compressive stress in *Y*-direction (MPa)**	−500	−212.6	3.16	−235.8
**Shear stress in *XY*-plane (MPa)**	118	2.574	45.91	6.431
**Von Mises stress of AL inserts (MPa)**	505	332.4	1.52	362.3

## Data Availability

Data sharing not applicable.

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
