# Peer review of "Design and Analysis of Solid Rocket Composite Motor Case Connector Using Finite Element Method"

_polymers, 2022, doi:10.3390/polym14132596_

Round 1

Reviewer 1 Report

Mechanical modelling of solid state fuel cartridge was performed by 3D numerical simulations for promising geometry and composite materials. it provides useful insights for future experimentation. 

one thing which can be improved, is estimation of structure performance under changing mass and resenance frequencies. this undermine stability. with the same spirit, the shock loading standard to detonation (Technologies 20197(4), 75; https://doi.org/10.3390/technologies7040075) is another limitation. can some insights in structure performance at those extreme loadings be anticipated?  

Reviewer 2 Report

The paper presents a FEM optimization of a CFRP SRMC connector. This evaluator has the following observations:

  • a general observation - there are plenty of EN language mistakes, including sentences or phrases that are difficult to follow or don't have a verb; please use an EN specialist to correct your paper

- in the Introduction section, please use references for your claims (e.g. the SRMC connector will affect blasting performance, same for the range of rocket, etc.) as otherwise they are not credible;

- can you please add a simple schematic (image) so the reader can understand better what the SRMC connector is (it was difficult for me to visualise it);

- please change CATIS to CATIA and add the owner - Dassault System's Catia TM;

- the literature review is biased towards the paper field - please also include paper related to CFRP in other applications and not only military.

- what is the maximum temperature for the connector? instead of just saying "more than 250 degrees" please use "up to a maximum of...", as otherwise seems fuzzy.

- Section 2 - please use a reference to the material used (manufacturer's);

- Section 3 - what is the first round of optimization? no details are provided; please include details about the process -as it is, it only seems a design change

- please explain in detail the choice of materials used and make the connection to the Journal you are trying to publish (polymers);

- Table 2 is unclear - what is the initial configuration?

- section 5.2 - please explain why you have chosen those particular loads and constraints, with references; also, a bit unclear whether you are discussing front or rear connectors, or both; please make the text more clear

- section 6 - authors are claiming that the results are in line with the "practical requests" - please use references to industrial standards or official regulations for this and compare to the traditional not improved versions to see the actual improvements

- The Conclusion section needs to be improved to accommodate most of the observations above; furthermore, please add further work and investigations for your research

Round 2

Reviewer 2 Report

Although you have addressed some of the issues raised by this evaluator, the paper still needs some work in order to make it more clear to the readers. Furthermore, the English style still require a lot of editing, as there are plenty of mistakes. References added are sometimes written in capitals 9both in the references section and in text). Literature review has not been completed as suggested. Forces applied in FEM are still not clear, although the authors have provided some explanations. Conclusion section has not be rewritten as suggested.
